# Patient-reported outcome measures for patients with meniscal tears: a systematic review of measurement properties and evaluation with the COSMIN checklist

Simon GF Abram, Robert Middleton, David J Beard, Andrew J Price, Sally Hopewell

Nuffield Department of Orthopaedics, Rheumatology and Musculoskeletal Sciences, University of Oxford, Oxford, UK

**Correspondence to**
Dr Simon GF Abram;
simon.abram@doctors.org.uk

## ABSTRACT

**Objective** Meniscal tears occur frequently in the population and the most common surgical treatment, arthroscopic partial meniscectomy, is performed in approximately two million cases worldwide each year. The purpose of this systematic review is to summarise and critically appraise the evidence for the use of patient-reported outcome measures (PROMs) in patients with meniscal tears.

**Design** A systematic review was undertaken. Data on reported measurement properties were extracted and the quality of the studies appraised according to Consensus-based Standards for the Selection of Health Measurement Instruments.

**Data sources** A search of MEDLINE, Embase, AMED and PsycINFO, unlimited by language or publication date (last search 20 February 2017).

**Eligibility criteria for selecting studies** Development and validation studies reporting the measurement properties of PROMs in patients with meniscal tears were included.

**Results** 11 studies and 10 PROMs were included. The overall quality of studies was poor. For measurement of symptoms and functional status, there is only very limited evidence supporting the selection of either the Lysholm Knee Scale, International Knee Documentation Committee Subjective Knee Form or the Dutch version of the Knee injury and Osteoarthritis Outcome Score. For measuring health-related quality of life, only limited evidence supports the selection of the Western Ontario Meniscal Evaluation Tool (WOMET). Of all the PROMs evaluated, WOMET has the strongest evidence for content validity.

**Conclusion** For patients with meniscal tears, there is poor quality and incomplete evidence regarding the validity of the currently available PROMs. Further research is required to ensure these PROMs truly reflect the symptoms, function and quality of life of patients with meniscal tears.

**PROSPERO registration number** CRD42017056847.

## Strengths and limitations of this study

► This is the first review of patient-reported outcome measures (PROMs) for patients with meniscal tears and the first to apply the Consensus-based Standards for the selection of health Measurement INstruments (COSMIN) checklist, which is a validated and accepted tool for the appraisal of study quality.

► Another strength of this review is the use of a validated, highly sensitive search strategy to identify relevant studies. A limitation, however, is that only studies specifically designed to appraise the measurement properties of PROMs were included. Trials and other clinical studies of patients with meniscal tears were not included as these studies are not designed to assess measurement properties, and the reporting of these properties would be highly unusual.

► Although the COSMIN checklist has acceptable inter-rater and intrarater properties, the scoring of some items is reliant on author's judgement. We performed pretesting to ensure scoring consistency, and review authors' scored studies independently. Nevertheless, it is feasible that another review team might score some items differently.

► For practical purposes, we chose to include tentative summary guidance regarding the selection of PROMs for use in the target population. It should be understood, however, that it would be reasonable to declare that the overall level of evidence for any of the PROMs is insufficient for a recommendation to be made.

## INTRODUCTION

The menisci are fibrocartilaginous structures within the knee joint that are important for load distribution and knee stability.[1 2] More than one-third of people over the age of 50 years without any radiographic evidence of osteoarthritis may develop a 'tear' of the meniscus, and over 70% of those with osteoarthritis will also have a torn meniscus.[3] These meniscal tears may be associated with significant knee pain and other symptoms, especially if the torn meniscal tissue interferes

with the normal articulation of the joint.[4] Meniscal tears are diagnosed and managed based on a combination of a review of symptoms, clinical examination and imaging findings.[5] Arthroscopic partial meniscectomy is a surgical procedure commonly used to treat symptomatic meniscal tears with approximately two million cases performed worldwide each year with combined costs of several billion US dollars.[6] A number of recent randomised controlled trials have been published challenging the effectiveness of arthroscopic partial meniscectomy.[7–11] Patient-reported outcome measures (PROMs) are critical to the interpretation of these trials, yet a wide array of different PROMs have been collected. This inconsistency leads to restricted comparisons between trials, and the best PROM for this population is unknown.[12 13]

PROMs are collected in a range of settings and are increasingly important in clinical practice. In orthopaedics, collecting PROM data is important in clinical practice to audit treatment outcomes and increasingly to demonstrate the cost-effectiveness of treatment.[14] With the rapid increase in usage, it is important to ensure that PROMs have formally validated measurement properties. Although generic PROMs enable the comparison of patients with different conditions, these PROMs may fail to capture important items in specific populations.[15] Ideally, a PROM should either be developed with condition-specific patient involvement or subsequently studied for validity in the population of interest.[16] Fundamentally, a PROM should comprehensively and consistently reflect the intended 'construct' to be measured in the population with the condition of interest, for example, health-related quality of life in patients with meniscal tears.[17]

There is a need for the selection of standardised 'core' PROMs for consistent use in clinical trials and the general clinical evaluation of patients with specific conditions.[13] A systematic review of the evidence is an important step in the selection of such a 'core outcome set' and may determine the need for further validation studies or even the development of a new PROM.[13] No systematic review has been published evaluating the measurement properties with the quality of evidence for the PROMs that are available for patients with meniscal tears. This is a barrier to the interpretation of previous research and to the design of future studies in these patients.

The purpose of this review is to report the measurement properties and evidence for the validity of all PROMs that have been evaluated in patients with meniscal tears.

## METHODS
This systemic review is reported based on the Preferred Reporting Items for Systematic Review and Meta-Analyses guidelines.[18] The protocol for this review was submitted to PROSPERO (CRD42017056847) on 20 February 2017.

### Study selection criteria
We included studies of adults with meniscal tears of the knee. Those studies with less than 50% of patients having a meniscal tear as the primary diagnosis (ie, without other significant knee pathology, for example, concomitant anterior cruciate ligament (ACL) rupture) were excluded unless the meniscal tear group was reported separately.

Studies administrating PROMs for the purpose of assessing measurement properties were included. PROMs using standard scoring methods, without clinician completed elements, measuring health-related quality of life, health status, symptoms including pain or functional status were included. Some studies included patients undergoing surgery (eg, arthroscopic partial meniscectomy and meniscal repair) as part of this assessment process, but the purpose of this review was not to assess the effectiveness of such interventions.

### Measurement properties
All PROM measurement properties reported by the included studies were evaluated. The primary measurement properties assessed were those within the reliability, validity and responsiveness domains. The secondary domains assessed were interpretability and generalisability. The Consensus-based Standards for the selection of health Measurement INstruments (COSMIN) definition of the domains and measurement properties follows below.[19]

#### Reliability
The reliability domain is a measure of how free a PROM is from measurement error.[19] The measurement properties within this domain are assessed by repeated collection of the PROM in a defined period when there has been no change in the patient's condition. Ideally, rather than assume the patient's condition is unchanged, a methodologically strong study will assess for change, for example, by administering a knee-specific global transition question on symptoms.
- Internal consistency is the degree of inter-relatedness among the PROM items.[19]
- Reliability is the proportion of total variance in the measurement which is because of true differences among patients.[19]
- Measurement error is the systematic and random error of a patient's score that is not attributed to true changes in the construct to be measured.[19]

#### Validity
The validity domain is the extent to which the PROM measures the 'construct' it purports to measure.[19]
- Content validity is the degree to which the content of a PROM is an adequate reflection of the construct to be measured.[19] The items should be comprehensive and relevant.
- Construct validity is the degree to which the scores of a PROM are consistent with hypotheses (eg, relationship of the score to that of other PROMs collected in the same group) based on the assumption the PROM validly measures the intended construct.[19]

- Structural validity is the degree to which the scores of a PROM are an adequate reflection of the dimensionality of the construct to be measured.[19]
- Hypothesis testing assumes that the PROM validly measures the construct of interest. Hypotheses are prepared a priori with regards to the correlation of the PROM with other relevant PROMs or domains of other PROMs. The magnitude and direction of the correlation should be stated in advance of testing.
- Cross-cultural validity is the degree to which the performance of the items on a translated or culturally adapted PROM are comparable with the performance of the original version of the PROM.[19]

### Responsiveness

Responsiveness is defined as the ability of a PROM to measure change over time in the construct to be measured.[19] It is important to note that in studies assessing the measurement properties of a PROM, responsiveness should be assessed against another valid PROM as for the assessment of construct validity. Measurement of effect size alone is not appropriate as this is a measure of the magnitude of the change and not the quality of the measurement.[20]

### Interpretability

Interpretability is defined as the degree to which it is possible to assign qualitative meaning to a PROM's quantitative score.[19] It is not considered a measurement property but is important when interpreting the findings from administration of a PROM in the context of a clinical condition. Interpretability includes an assessment of minimal important change (MIC) and floor and ceiling effects. In general, floor and ceiling effects <15% are considered acceptable, although some authors have argued the threshold should be set at <30%.[21 22] A high floor or ceiling effect suggests that items at the lower or upper end are missing from a question item, domain or the PROM overall.

### Generalisability

Generalisability is an assessment of external validity: the extent to which the findings on the measurement properties of a PROM may be considered relevant to a population or construct of interest. For example, a study of the measurement properties of a PROM in a population with advanced knee osteoarthritis cannot be generalised to athletes with knee ligament injury without further study in the target population. In this review, the population of patients involved in the original development of each PROM is determined, and the inclusion and exclusion criteria of all studies reporting measurement properties of the included PROMs are reported to highlight any heterogeneity. The generalisability of findings to the population of patients with meniscal tears is considered.

### Search strategy

We performed a search of MEDLINE, Embase, AMED and PsycINFO, unlimited by language or publication date. The search was based on a validated search filter designed to be highly sensitive in identifying all studies of measurement properties.[23] Full details of the search are available in online supplementary file 1. The final search was performed on 20 February 2017, following submission of the protocol to PROSPERO. A review of study citations was performed to further increase the sensitivity of the search strategy.

### Selection of studies

The title and abstract of all records retrieved by the search was independently reviewed by two authors against the inclusion and exclusion criteria (SGFA and RM). Any disagreement was resolved with review of the full text publication and discussion. Referral to a third author (SH) was not required for agreement. The original PROM development article was retrieved for all PROMs identified, for example, where a PROM was developed for a condition other than meniscal pathology and subsequently tested in a population with meniscal tears.

### Data extraction: measurement properties and assessing the quality of studies

Data extraction was performed by two authors (SGFA and RM) and any disagreement was resolved in consultation with a third author (SH). The following was extracted from each publication: the PROM, the intended construct for measurement, measurement properties, administration method, study population and diagnosis, number of patients, patient demographics, country, language and setting and method of administration (eg, postal and online).

The quality of each included study was assessed by two reviewers (SGFA and RM) using the COSMIN appraisal checklist.[24] When reviewing a study of a PROM, it is necessary to consider a combination of the reported measurement properties, the patient population and the quality of the study methodology. To help overcome some of the difficulties in evaluating the quality of PROMs, COSMIN was published in 2010.[20 25] COSMIN contains rules for grading overall methodological quality of studies performed into the measurement properties of PROMs. These consensus standards are regularly reviewed and revised based on the latest evidence and research. COSMIN initially separated standards into boxes including a series of binary methodological ratings. The scoring methodology was subsequently revised to a four-level (excellent/good/fair/poor) rating system in 2012.[24] Each measurement property is assessed by a box containing 5–18 questions scored on this scale according to defined COSMIN criteria. A system of 'worst score counts' applies for each box, that is, if one question in the box is scored as poor, the overall quality of the evidence for that measurement property is determined to be poor.

**Table 1** Overall levels of evidence for the quality of the measurement property.[26 27] The quality of the evidence for the measurement property for each PROM, considering the quality criteria for each measurement property (online supplementary file 2), the methodology of each study reporting the measurement property (table 4) and the number of studies reporting the measurement property including consistency of findings

| Level of evidence | Rating | Quality criteria |
|---|---|---|
| Strong | +++ or − − − | Consistent findings (positive or negative) in multiple studies of good methodological quality OR in one study of excellent methodological quality |
| Moderate | ++ or − − | Consistent findings (positive or negative) in multiple studies of fair methodological quality OR in one study of good methodological quality |
| Limited | + or − | One study of fair methodological quality (positive or negative) |
| Conflicting | +/− | Conflicting results |
| Unknown | ? | Only studies of poor methodological quality |

+=positive rating, ?=indeterminate rating, −=negative rating.
PROM, patient-reported outcome measure.

## Data synthesis

Data synthesis was performed by SGFA and checked by RM. For each included PROM, a summary of the features of the PROM is presented including details of the original development process, the development population and target construct to be measured.

For each PROM, a rating (positive, negative or indeterminate) for the measurement properties reported in the study was first determined based on consensus standards described in online supplementary file 2.[22] This assessment was then combined with an overall quality of evidence assessment, which was adapted for COSMIN from the work of the Cochrane Back Review Group (table 1).[26 27] For example, one good quality study reporting positive measurement properties (eg, internal consistent with Cronbach's alpha ≥0.70) results in an overall rating of 'moderate' (++). Where the quality of study methodology on a measurement property is rated 'poor', the overall rating of the measurement property is always rated 'indeterminate', irrespective of the number of such studies and whether the reported measurement property itself would otherwise be considered positively. These standards are designed to ensure reported measurement properties are interpreted in the context of study quality and overall reliability.

## RESULTS

### Selection of studies

The search strategy identified 1321 unique articles for screening. After screening, 34 full-text articles were retrieved of which 11 met the inclusion criteria for this review. Figure 1 summarises the study selection process. The 11 studies reported measurement properties for 10 PROMs.

### Study characteristics

The characteristics of the included studies are shown in table 2. The mean age of patients included in the studies ranged from 38 to 53 years. The proportion of female patients included ranged from 14% to 64%. Inclusion and exclusion criteria were inconsistent with regards to age, symptoms, investigations and treatment (table 2). The development and features of the included PROMs are summarised below and in table 3.

### Quality of the included studies

In total, the 11 studies reported 93 measurement properties for the 10 PROMs. The COSMIN methodology rating for 49 of these (53%) was poor. Many measurement properties were not reported, and there was inconsistent reporting between studies (table 4).

### Quality of PROMs

Interpretability factors including floor and ceiling effects are summarised in table 5. The overall level of evidence for the measurement properties of each PROM is summarised in table 6. This combines the rating of the reported measurement property using the consensus criteria available in online supplementary appendix 2 with the COSMIN scoring and the number of studies per PROM (as described in table 1).

Of the 10 PROMs identified, five intended to measure symptoms and functional status, four health-related quality of life and one activity level.

### Symptoms and functional status
#### Hughston
The Hughston Clinic Questionnaire was developed in 1991 as a knee-specific rather than disease-specific outcome measure.[28] It includes questions on symptoms, functional status and sports activity, and patients were not involved in the development of the questions. Only one study has evaluated use of the Hughston questionnaire in patients with meniscal tears.[29] Content validity was rated poor as patients were not involved in the original development, and content validity has not been subsequently assessed in patients with meniscal tears. In patients with meniscal tears, there was moderate negative evidence against construct validity based on hypothesis testing,

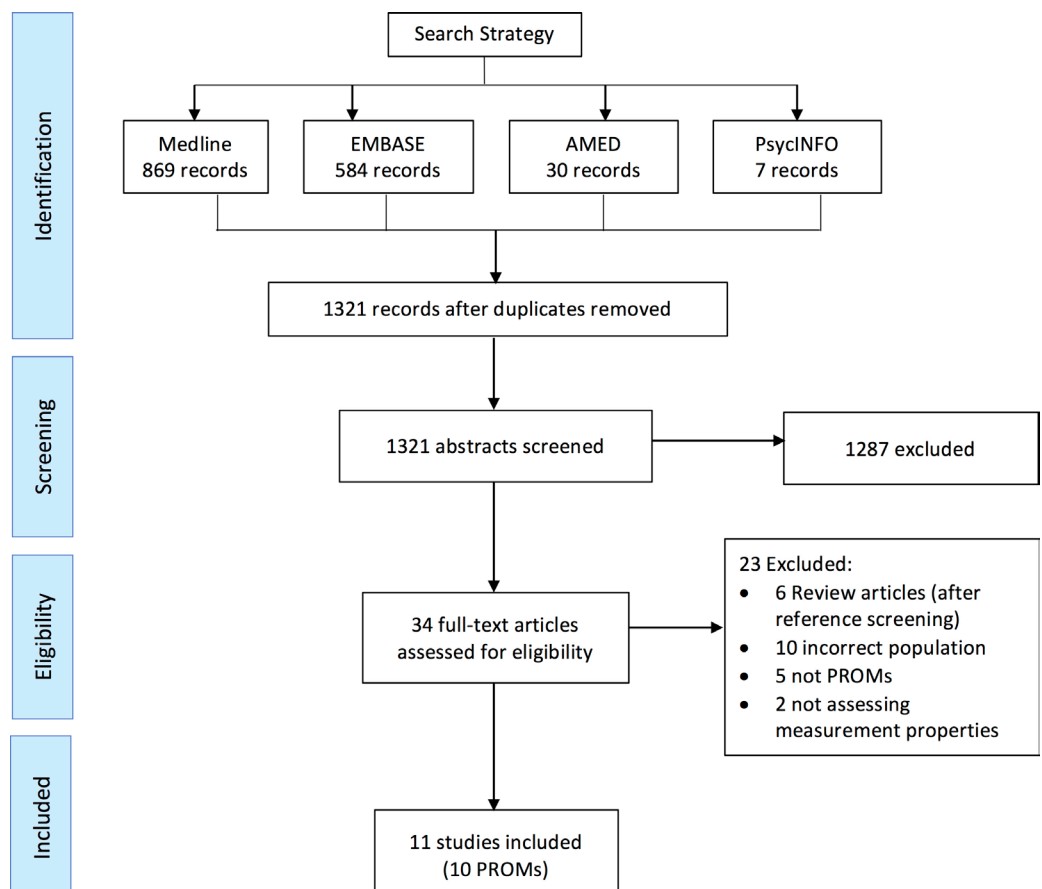

**Figure 1** PRISMA flow diagram. Overview of study selection. Full search strategy may be found in online supplementary file 1. PRISMA, Preferred Reporting Items for Systematic Review and Meta-Analyses; PROMs, patient-reported outcome measures.

and all other measurement properties were either not reported or indeterminate due to poor study design or reporting (table 6).

### IKDC

The International Knee Documentation Committee (IKDC) Subjective Knee Form was developed in 2001 as a knee-specific rather than disease-specific outcome measure.[30] It includes question domains on symptoms, functional status and sports activity, and patients were not involved in the development of the questions. Two studies have evaluated use of the IKDC score in patients with meniscal tears.[31 32] In English, there is limited positive evidence for reliability and construct validity based on hypothesis testing.[31] In Dutch, there is moderate positive evidence for reliability and construct validity based on hypothesis testing but limited negative evidence against structural validity.[32] In both studies, all other measurement properties were either not reported or indeterminate (table 6). For the English version, although no floor or ceiling effect was detected for the overall score, unacceptable floor effects were reported for nine items and unacceptable ceiling effects in five items (table 5).

### KOOS

The Knee injury and Outcome Osteoarthritis Score (KOOS) was developed in 1998 as a knee injury-specific

outcome measure for patients at risk of developing osteoarthritis.[33] It includes question domains on symptoms, functional status, sports activity and quality of life. Patients with ACL or meniscal injuries were included in the development process. The KOOS includes the Western Ontario McMaster Osteoarthritis Index (WOMAC) osteoarthritis score in full, and the WOMAC may therefore be calculated from the KOOS. The KOOS has been studied in Dutch and Swedish for patients with meniscal tears; no study has evaluated the English version of KOOS in this population.[32 34] There is moderate positive evidence for reliability and construct validity from hypothesis testing of the Dutch version.[32] For the Swedish version, there is limited positive evidence for reliability and construct validity based on hypothesis testing.[34] For both the Dutch and Swedish versions, content validity and all other measurement properties were either rated indeterminate or were not reported (table 6).

### Lysholm

The Lysholm Knee Scale was developed in 1982 and modified in 1985 as a disease-specific outcome measure for patients with knee ligament injury.[35 36] The Lysholm Knee Score was originally designed to be completed by clinicians and developed without patient involvement. One study has evaluated the use of Lysholm in

**Table 2** Characteristics of the included studies

| Study (year) | Instrument(s) | Country (language) | Population (inclusion and exclusion criteria) | N | Mean age (SD, range) | Female: Male |
|---|---|---|---|---|---|---|
| Goodwin et al (2011)[29] | Hughston EQ-5D SF-6D | UK (English) | Inclusion: patients previously undergoing arthroscopic partial meniscectomy. | 84 | 38 (SD 8, 21–58) | 14: 86% |
| Crawford et al (2007)[31] | IKDC | USA (English) | Inclusion: patient with 'meniscal pathology requiring treatment' and completed IKDC questionnaire. Exclusion: patients with ligament pathology or a chondral defect greater than Outerbridge grade 2 | Groups: A: 31 B: 264 C: 50 D: 50 | 48 (18–81) | 29: 71% |
| van de Graaf et al (2014)[32] | IKDC KOOS WOMAC | Netherlands (Dutch) | Inclusion: age >18 years, knowledge of Dutch language, either on waiting list for meniscal surgery or between 6 weeks and 6 months after meniscal surgery. Exclusion: received arthroplasty in either knee or previous ACL surgery on the knee of interest. | 75 | 48.8 (35–62) | 50: 50% |
| Roos et al (1998)[34] | KOOS | Sweden (Swedish) | Inclusion: patient waiting for knee arthroscopy for either meniscal lesion, ACL injury or tibiofemoral cartilage damage. (54% meniscal tear, 20% ACL+meniscal tear, 13% ACL, 13% isolated chondral damage) Exclusion: multiple joint involvement, other diagnosis. | 142 | 39.7 (14–75) | 22: 78% |
| Garratt et al (2008)[39] | KQoL-26 | UK (English) | Inclusion: patients aged 18–55 years, referred to hospital clinic with suspected meniscus or knee ligament pathology. (67% meniscal tear, 30% ACL, 3% other) Exclusion: Requiring urgent referral, non-traumatic arthropathy, chronic knee instability, previous same knee surgery (except diagnostic arthroscopy) | 323 | 47 (14.3) | 44: 56% |
| Briggs et al (2006)[21] | Lysholm Tegner | USA (English) | Inclusion: Patient previously undergoing surgery for meniscal lesion or waiting list for meniscal surgery. | Groups: A: 122 B: 191 C: 477 | 40 (13–81) | 32: 68% |
| Kirkley et al (2007)[41] | WOMET | Canada (English) | Inclusion: patients with 'meniscal symptoms (swelling, catching, locking)' and MRI suggestive of meniscal pathology. | Groups: A: 31 B: 36 C: 34 D: 69 | Not reported | Not reported |
| Sihvonen et al (2012)[59] | WOMET | Finland (Finnish) | Inclusion: patients with arthroscopically verified degenerative meniscal tear and no previous knee trauma. Exclusion: trauma, bilateral arthroscopy, reoperation within 6 months | Groups: A: 485 B: 385 C: 100 D: 40 | 53 (18–81) | 45: 55% |
| Celik et al (2015)[60] | WOMET | Turkey (Turkish) | Inclusion: age >16 years, presence of meniscal tear or previous meniscal repair or resection, complete questionnaires. Exclusion: ligament injury, 'articular cartilage damage causing instability', inability to complete the form due to cognitive impairment. | 96 | 43.6 (23–71) | 64: 36% |
| Tong et al (2016)[61] | WOMET | China (Chinese) | Inclusion: patients with meniscal pathology who underwent arthroscopic surgery for meniscal repair or resection. Age >18 years, able to read and speak Chinese. Exclusion: ligament injuries, history of leg surgery, infection, tumours, rheumatological disease, neurological or musculoskeletal disorders. | 121 | 41.2 (14.3) | 57: 43% |
| van der Wal et al (2016)[51] | WOMET | Netherlands (Dutch) | Inclusion: patients with MRI confirmed, symptomatic, meniscal tear. Age 18–70 years, understanding of Dutch language. Exclusion: concomitant ligament injury, previous ligament injury with instability, previous knee surgery, chondral defect greater than Outerbridge grade 2 on MRI or during surgery, inability to participate due to cognitive impairment. | 86 | Median 52 (IQR 43–60) | 41: 59% |

ACL, anterior cruciate ligament; EQ-5D, EuroQoL-5 dimension; IKDC, International Knee Documentation Committee; KOOS, Knee injury and Osteoarthritis Outcome Score; KQoL-26, 26-item Knee Quality of Life; SF-6D, Short Form-6 dimensions; WOMAC, Western Ontario McMaster Osteoarthritis Index; WOMET, Western Ontario Meniscal Evaluation Tool.

**Table 3** Characteristics of the included PROMs

| Instrument | Year of development | Original language | Intended construct and domains | Number of questions | Target or development population | Patients involved in development? |
|---|---|---|---|---|---|---|
| **Symptoms and functional status** | | | | | | |
| Hughston | 1991[29] | English | Knee-specific symptoms, functional status and sports activity. No subdomains | 28 questions | 'Patients who had undergone knee surgery that varied from arthroscopy to total arthroplasty'.[28] | No |
| IKDC | 2001[30] | English | Knee-specific symptoms, functional status and sports activity. 1. Symptoms 2. Sports activities 3. Function | 18 questions | 'A knee-specific, rather than a disease-specific, measure of symptoms, function, and sports activity'.[30] | No |
| KOOS | 1998[33] | English | Knee injury-specific symptoms, functional status, sports activity and quality of life (QoL). 1. Symptoms and stiffness 2. Pain 3. Activities of daily living 4. Function in sports and recreation 5. Knee-related QoL | 42 questions | Patients with knee injury (ACL or meniscus injury) at risk of developing osteoarthritis. | Yes |
| Lysholm | 1982[35]/1985[36] | English | Disease-specific (knee ligament) symptoms and functional status No subdomains | Eight questions | 'A scoring scale for knee ligament surgery follow-up emphasizing evaluation of symptoms of instability'.[35] | No |
| WOMAC | 1982[37] | English | Disease-specific (osteoarthritis of hip or knee) symptoms, functional status 1. Pain 2. Stiffness 3. Function and daily activities | 24 questions | 'Outcomes of anti-rheumatic drug therapy in patients with osteoarthritis of the hip or knee'.[62] | Yes |
| **Health-related quality of life** | | | | | | |
| EQ-5D | 1990[38] | English, Dutch, Finnish, Norwegian, Swedish | General population health-related QoL 1. Mobility 2. Self-care 3. Usual activities 4. Pain/discomfort 5. Anxiety/depression | Six questions | General tool for describing and valuing health-related QoL—items developed and valued after questioning large samples of randomly selected adults. | Yes |
| KQoL-26 | 2008[39] | English | Disease-specific (knee ligament or meniscus) health-related QoL 1. Physical functioning 2. Activity limitations 3. Emotional functioning | 26 questions | 'Patients with a suspected ligamentous or meniscal injury of the knee'.[39] | Yes |
| SF-6D | 2004[40] | English | General population health-related QoL 1. Physical functioning 2. Role limitation 3. Social functioning 4. Pain 5. Mental health 6. Vitality | Six questions | Derived from SF-36 or SF-12. A general, preference based classification for describing health-related QoL. | Yes |
| WOMET | 2007[41] | English | Disease-specific (meniscus) health-related QoL Physical symptoms 1. Sports/recreation/work/lifestyle 2. Emotions | 16 questions | 'Patients with meniscal symptomology (swelling, catching, locking) and in whom magnetic resonance imaging had suggested meniscal pathology'.[41] | Yes |
| **Activity level** | | | | | | |
| Tegner | 1985[36] | English | Disease-specific (knee ligament) symptoms and functional status No subdomains | One question | Patients with ACL injury diagnosed by clinical examination under anaesthesia and confirmed by arthroscopy or arthrotomy. | No |

ACL, anterior cruciate ligament; EQ-5D, EuroQoL-5 dimension; IKDC, International Knee Documentation Committee; KOOS, Knee injury and Osteoarthritis Outcome Score; KQoL-26, 26-item Knee Quality of Life; PROM, patient-reported outcome measure; SF-6D, Short Form-6 dimensions; WOMAC, Western Ontario McMaster Osteoarthritis Index; WOMET, Western Ontario Meniscal Evaluation Tool.

**Table 4** Quality of each study per PROM and measurement property (COSMIN rating)

| Instrument and study | Internal consistency | Reliability | Measurement error | Content validity | Structural validity | Hypothesis testing | Cross-cultural validity | Responsiveness |
|---|---|---|---|---|---|---|---|---|
| **Symptoms and functional status** | | | | | | | | |
| Hughston | | | | | | | | |
| Goodwin et al[29] | Poor | na | na | Poor | na | Good | na | Poor |
| IKDC | | | | | | | | |
| Crawford et al[31] | Poor | Fair | Fair | Poor | na | Fair | na | Poor |
| van de Graaf et al[32] | Poor | Good | Good | Poor | Fair | Good | Poor | na |
| KOOS | | | | | | | | |
| Roos et al[34] | Poor | Fair | na | Poor | Poor | Fair | Poor | Poor |
| van de Graaf et al[32] | Poor | Good | Good | Poor | Poor | Good | Poor | na |
| Lysholm | | | | | | | | |
| Briggs et al[21] | Poor | Fair | Fair | Poor | na | Fair | na | Poor |
| WOMAC | | | | | | | | |
| van de Graaf et al[32] | Poor | Good | Good | Poor | Poor | Good | Poor | na |
| **Health-related quality of life** | | | | | | | | |
| EQ-5D | | | | | | | | |
| Goodwin et al[29] | Poor | na | na | Poor | na | Good | na | Poor |
| KQoL–26 | | | | | | | | |
| Garratt et al[39] | Fair | Fair | na | Fair | Fair | Fair | na | Poor |
| SF-6D | | | | | | | | |
| Goodwin et al[29] | Poor | na | na | Poor | na | Good | na | Poor |
| WOMET | | | | | | | | |
| Kirkley et al[41] | Poor | Fair | na | Excellent | na | Fair | na | Fair |
| Sihvonen et al[59] | Poor | Poor | na | Poor | na | Fair | Poor | Poor |
| Celik et al[60] | Poor | Good | Good | Poor | na | Good | Poor | na |
| Tong et al[61] | Poor | Good | na | Poor | na | Good | Poor | Poor |
| van der Wal[51] | Poor | Good | Good | Good | na | Good | Poor | Good |
| **Activity level** | | | | | | | | |
| Tegner | | | | | | | | |
| Briggs et al[21] | na | Fair | Fair | Poor | na | Fair | na | Poor |

COSMIN, Consensus-based Standards for the selection of health Measurement INstruments; EQ-5D, EuroQoL-5 dimension; IKDC, International Knee Documentation Committee; KOOS, Knee injury and Osteoarthritis Outcome Score; KQoL-26, 26-item Knee Quality of Life; PROM, patient-reported outcome measure; SF-6D, Short Form-6 dimensions; WOMAC, Western Ontario McMaster Osteoarthritis Index; WOMET, Western Ontario Meniscal Evaluation Tool.

**Table 5** Interpretability including missing items, response rate and floor and ceiling effects

| Instrument and study | Administration | Missing responses (%) | Missing items (%) | Overall % lowest possible total score (floor) | Overall % highest possible score (ceiling) | Items or Domains with >15% responses with lowest score (floor) | Items or domains >15% highest possible score (ceiling) | MIC |
|---|---|---|---|---|---|---|---|---|
| **Symptoms and functional status** | | | | | | | | |
| Hughston | | | | | | | | |
| Goodwin et al[29] | Clinic | Not reported | Not reported | 0 | 0 | Not reported | Not reported | Not reported |
| IKDC | | | | | | | | |
| Crawford et al[31] | Clinic/postal | Not reported | Not reported | 0 | 0 | ▲ Activity pain ▲ Pain last 4 weeks ▲ Pain severity ▲ Catching ▲ Kneeling ▲ Sitting ▲ Running ▲ Jumping ▲ Stopping | ▲ Swelling ▲ Catching ▲ Climb stairs ▲ Sitting ▲ Rising | Not reported |
| van de Graaf et al[32] | Online/postal | Unclear | 0 | 0 | 0 | Nil | Nil | Not reported |
| KOOS | | | | | | | | |
| Roos et al[34] | Postal | 7.2 | 0.8 | Not reported | Not reported | Nil | Nil | Not reported |
| van de Graaf et al[32] | Online/postal | Unclear | 0 | 0 | 3 | Nil | Nil | Not reported |
| Lysholm | | | | | | | | |
| Briggs et al[21] | Clinic | Not reported | Not reported | 0% | 0.5% | ▲ Squatting ▲ Pain | ▲ Swelling ▲ Instability ▲ Support ▲ Limp ▲ Locking | Not reported |
| WOMAC | | | | | | | | |
| van de Graaf et al[32] | Online/Postal | Unclear | 0 | 0 | 6 | Nil | Nil | Not reported |
| **Health-related quality of life** | | | | | | | | |
| EQ-5D | | | | | | | | |
| Goodwin et al[29] | Clinic | Not reported | Not reported | 4 | 1 | Not reported | Not reported | Not reported |
| KQoL-26 | h31 | | | | | | | |
| Garratt et al[39] | Postal | 41 | 14.9 | Not reported | Not reported | ▲ Avoiding turning, twisting, or sideways movements ▲ * | ▲ Staying seated for 15 min ▲ * | Not reported |
| SF-6D | | | | | | | | |
| Goodwin et al[29] | Clinic | Not reported | Not reported | 0 | 0 | Not reported | Not reported | Not reported |
| WOMET | | | | | | | | |
| Kirkley et al[41] | Clinic | Not reported | Not reported | 5.7 | 1.7 | Not reported | Not reported | Not reported |
| Silvonen et al[59] | Unclear | 16 | 7.5 | 0 | 0 | Numbness | Nil | Not reported |

Continued

**Table 5** Continued

| Instrument and study | Administration | Missing responses (%) | Missing items (%) | Overall % lowest possible total score (floor) | Overall % highest possible score (ceiling) | Items or Domains with >15% responses with lowest score (floor) | Items or domains >15% highest possible score (ceiling) | MIC |
|---|---|---|---|---|---|---|---|---|
| Celik et al[60] | Unclear | Not reported | Not reported | 0 | 0 | ▲ Numbness<br>▲ Swelling | ▲ Consciousness<br>▲ Activities<br>▲ Specific skills<br>▲ Squatting<br>▲ Fear injury<br>▲ Concern about future of knee<br>▲ Frustration | Not reported |
| Tong et al[61] | Unclear | Not reported | 0 | 0 | 0 | Nil | Nil | Not reported |
| van der Wal et al[51] | Clinic | 0 | <1 | 0 | 0 | ▲ Numbness<br>▲ Swelling | Nil | 14.7 |
| **Activity level** | | | | | | | | |
| Tegner | | | | | | | | |
| Briggs et al[21] | Clinic | Not reported | Not reported | 2.5 | 2.5 | na | na | Not reported |

*Other domains not reported.
EQ-5D, EuroQoL-5 dimension; IKDC, International Knee Documentation Committee; KOOS, Knee injury and Osteoarthritis Outcome Score; KQoL-26, 26-item Knee Quality of Life; MIC, minimalimportant change; SF-6D, Short Form-6 dimensions; WOMAC, Western Ontario McMaster Osteoarthritis Index; WOMET, Western Ontario Meniscal Evaluation Tool.

English-speaking patients with meniscal tears.[21] There is limited positive evidence for reliability and construct validity based on hypothesis testing. Content validity and all other measurement properties are either indeterminate or were not reported (table 6). There was no floor or ceiling effect for the Lysholm score overall; however, an unacceptable floor effect was detected for two items and unacceptable ceiling effects for five items (table 5).

### Western Ontario McMaster Osteoarthritis Index
The WOMAC was developed in 1982 as a disease-specific outcome measure for patients with osteoarthritis of the hip or knee.[37] The WOMAC includes question domains for pain, stiffness and functional status, and patients with osteoarthritis were involved in the development of the questions. The WOMAC is incorporated in its entirety in the KOOS (see above). One study has evaluated the Dutch version of WOMAC in patients with meniscal tears.[32] In these patients, there is moderate positive evidence for reliability and construct validity (hypothesis testing). No floor or ceiling effects were detected. Content validity and all other measurement properties are either indeterminate or were not reported (table 6).

### Health-related quality of life
#### EuroQoL-5 dimension (EQ-5D)
EQ-5D is a generic measure of health-related quality of life developed in 1990.[38] It was developed with patient involvement and includes question domains on mobility, self care, usual activities, pain, and anxiety or depression. One study has evaluated the English EQ-5D in patients with meniscal tears.[29] In this population, there is moderate positive evidence for construct validity based on hypothesis testing. All other measurement properties are either indeterminate or were not reported.

#### Twenty-six-item Knee Quality of Life (KQoL-26)
The KQoL-26 26-item questionnaire was developed in 2008, in English, as a disease-specific health-related quality of life measure for patients with suspected ligamentous or meniscal injury of the knee.[39] In the study population, 67% of patients had a meniscal tear and there is limited positive evidence for internal consistency, reliability, content validity, and construct validity (hypothesis testing and structural validity). Administered by post, an overall response rate of 59% was reported with 14.9% missing items.[39] Floor and ceiling effects were poorly reported with at least one question having an unacceptable floor effect and one an unacceptable ceiling effect (table 5).

#### Short Form-6 dimensions (SF-6D)
The Short Form-6 dimensions (SF-6D) generic health-related quality of life measure is derived from the 36-Item Short Form Survey (SF-36) or 12-item Short Form Survey (SF-12) and was developed in 2004.[40] It was developed with patient involvement and contains six questions domains: physical functioning, role limitation, social functioning, pain, mental health and vitality. One study has evaluated the English version of SF-6D in patients

**Table 6** Overall rating of measurement properties and level of evidence for each PROM. See table 1 for a summary of the rating methodology

| Instrument | Internal consistency | Reliability | Measurement error | Content validity | Structural validity | Hypothesis testing | Cross-cultural validity | Responsiveness |
|---|---|---|---|---|---|---|---|---|
| **Symptoms and functional status** | | | | | | | | |
| Hughston | | | | | | | | |
| English[29] | ? | na | na | ? | na | − − | na | ? |
| IKDC | | | | | | | | |
| English[31] | ? | + | ? | ? | na | + | na | ? |
| Dutch[32] | ? | ++ | ? | ? | − | ++ | ? | na |
| KOOS | | | | | | | | |
| Dutch[32] | ? | ++ | ? | ? | ? | ++ | ? | na |
| Swedish[34] | ? | + | na | ? | ? | + | ? | ? |
| Lysholm | | | | | | | | |
| English[21] | ? | + | ? | ? | na | + | na | ? |
| WOMAC | | | | | | | | |
| Dutch[32] | ? | ++ | ? | ? | ? | ++ | ? | na |
| **Health-related quality of life** | | | | | | | | |
| EQ-5D | | | | | | | | |
| English[29] | ? | na | na | ? | na | ++ | na | ? |
| KQoL–26 | | | | | | | | |
| English[39] | + | + | na | + | + | + | na | ? |
| SF-6D | | | | | | | | |
| English[29] | ? | na | na | ? | na | ++ | na | ? |
| WOMET | | | | | | | | |
| English[41] | ? | + | na | +++ | na | + | na | + |
| Chinese[61] | ? | ++ | na | ? | na | ++ | ? | ? |
| Dutch[51] | ? | ++ | − − | ++ | na | ++ | ? | ++ |
| Finnish[59] | ? | ? | na | ? | na | + | ? | ? |
| Turkish[60] | ? | ++ | ? | ? | na | ++ | ? | na |
| **Activity level** | | | | | | | | |
| Tegner | | | | | | | | |
| English[21] | na | + | ? | ? | na | + | na | ? |

EQ-5D, EuroQoL–5 dimension; IKDC, International Knee Documentation Committee; KOOS, Knee injury and Osteoarthritis Outcome Score; KQoL–26, 26-item Knee Quality of Life; PROM, patient-reported outcome measure; SF-6D, Short Form-6 dimensions; WOMAC, Western Ontario McMaster Osteoarthritis Index; WOMET, Western Ontario Meniscal Evaluation Tool.

with meniscal tears.[29] There is moderate positive evidence for construct validity based on hypothesis testing, but all other measurement properties are indeterminate or were not reported.

### Western Ontario Meniscal Evaluation Tool (WOMET)

The WOMET is a meniscal tear disease-specific quality of life measure developed in 2007.[41] Patients with meniscal tears were involved throughout the development process, although the authors reported that the same patients were 'admittedly heterogeneous with respect to the incidence of coexisting knee pathology such as chondral damage or ligament injury'.[41] The WOMET has been evaluated in English, Chinese, Dutch, Finnish and Turkish. There is strong positive evidence for content validity in the English version and moderate positive evidence in the Dutch version. There is limited positive evidence for reliability, construct validity (hypothesis testing) and responsiveness of the English version.[41] Measurement error was only reported for the Dutch version of WOMET and in this case it was concerning that the MIC for the PROM was found to be less than the smallest detectable change (SDC). A summary of the level of evidence for the measurement properties in all languages is shown in table 6. Although the overall score does not exhibit floor or ceiling effects, unacceptable levels were reported for several items (table 5).

### Activity level
#### Tegner
The Tegner Activity Scale was developed in 1985 for patients with ACL injury.[36] Patients were not involved in the development of the scale. One study has evaluated use of the scale in patients with meniscal tears.[21] In this population, there is limited positive evidence for reliability and construct validity based on hypothesis testing. All other measurement properties were either not reported or indeterminate.

### DISCUSSION
This review identified 11 studies evaluating 10 PROMs in patients with meniscal tears: five PROMS measuring symptoms and functional status, four PROMs measuring health-related quality of life and one for activity level. Unfortunately, the findings of the studies were limited by poor methodology and incomplete reporting of measurement properties.

One previous review has been published summarising reported measurement properties of a range of PROMs in studies of patients with any knee condition.[42] In this previous review, WOMET was broadly recommended for use in patients with meniscal injuries without distinguishing the intended health-related quality of life construct from others or assessing the quality of the studies.[42] Ours is the first systematic review of PROMs for patients with meniscal tears and the first to evaluate and report the quality of study methodology. In orthopaedics

and sports medicine, systematic reviews of PROMs applying the COSMIN appraisal checklist are established and have been published for patient populations including those with hip and knee osteoarthritis, hip and groin disability, patellofemoral pain, distal radius fractures, shoulder pain and undergoing hip arthroscopy.[43–50]

For studies included in this review, the COSMIN methodology rating was poor for just over half (53%) of reported measurement properties. Internal consistency was rated poor in all but 1 of the 11 studies. A key reason for this was the failure of most studies to perform factor analysis to assess the structural validity of PROMs. Internal consistency is an assessment of the inter-relatedness of the items measuring the same underlying construct, that is, the PROM or subdomain should be 'unidimensional' for the construct to be measured. Factor analysis is a technique that may be used to determine whether a PROM or subdomain is 'unidimensional'. Without this assessment of structural validity, there can be no clear interpretation of internal consistency statistics.[20]

Cross-cultural validity and responsiveness were also particularly poorly evaluated. Regarding responsiveness, frequently studies reported only an effect size for the studied PROM. Effect size alone is a measure of the magnitude of a change score and not the quality of the measurement and is therefore insufficient to assess this measurement property.[20] Responsiveness refers to the validity of a change score and should be assessed with, for example, hypothesis testing against the change score of another related PROM, analogous to the assessment of construct validity.

Measurement error was poorly reported in the included studies, and the MIC was calculated for only one of the PROMS—the Dutch version of WOMET.[51] It was concerning that in this case the MIC was found to be less than the SDC due to measurement error. Failure to determine and report this information affects the ability of researchers to design high-quality prospective studies and limits interpretation of previous work.

Evidence for the content validity of the available PROMs was limited. Only the KQoL-26 and WOMET were developed with involvement from patients with meniscal tears. Overall, there was heterogeneity in the population of the patients recruited to the included studies as shown in table 2. Although most patients in the included studies had meniscal tears as their primary diagnosis, many also had a diagnosis of ligament injury or chondral damage. This reflects the heterogeneity of patients with meniscal tears in general, ranging from the isolated traumatic tear in a young athlete without osteoarthritis to atraumatic tears in older patients with osteoarthritis. Meniscal tears are not always symptomatic, and given the association with osteoarthritis, the distinction between the onset of meniscal pain and osteoarthritic pain is often unclear.[3 52] No single patient factor is sufficient in isolation.[5] For example, the degenerative meniscus will be more susceptible to tearing following knee trauma than a normal meniscus, and no difference

in symptom profile or treatment response has been demonstrated based on the mechanism of symptom onset.[53 54]

For studies in this review, there was significant variation in the methods used to identify patients. The latest guidance states specific types of meniscal tears should be identified on MRI and related to symptoms and other findings before any surgical intervention is recommended.[5] Several studies included only patients with meniscal tears verified by previous arthroscopic surgery, whereas others verified meniscal tears were visible on MRI. For all the included studies, it was unclear how patients were identified to have symptoms correlating with a meniscal tear rather than other pathology such as osteoarthritis. Identifying patients with symptoms that definitely originate from the meniscus is challenging for both clinicians and researchers.

The mean age range of patients included in the studies was 38–53 years, and therefore the generalisability of the findings to other age groups is unclear. It is highly likely that the symptom profile and expectations of younger, active patients sustaining a tear to a normal meniscus in an otherwise normal knee will be different to the study patients with predominantly degenerative meniscal tears and underlying osteoarthritis. This has not yet been evaluated.

### Strengths and limitations

One strength of this review is the use of a validated, highly sensitive search strategy to identify relevant studies.[23] A limitation, however, is that only studies specifically designed to appraise the measurement properties of PROMs were included. Trials and other clinical studies of patients with meniscal tears were not included as these studies are not designed to assess measurement properties, and the reporting of these properties would be highly unusual. For the same reason, clinical trial registries were not searched for ongoing studies.

This is the first review of PROMs for patients with meniscal tears and the first to apply the COSMIN checklist, which is a validated and accepted tool for the appraisal of study quality. Although it has been shown to have acceptable inter-rater and intrarater properties, the scoring of some items is reliant on author judgement.[55] We performed pretesting to ensure scoring consistency and review authors scored studies independently with any disagreement being settled by consensus or discussion with a third author. Nevertheless, it is feasible that another review team might score some items differently.

For practical purposes, we chose to include tentative summary guidance regarding the selection of PROMs for use in the target population. It should be understood, however, that it would be reasonable to declare that the overall level of evidence for any of the PROMs is insufficient for a recommendation to be made. Due to the study population limitations discussed earlier, the generalisability of the summary findings may also be challenged.

### Implications for practice

Currently, although a wide range of PROMs are available for patients with knee conditions, the PROMs that have been tested in patients with meniscal tears all lack data on a large proportion of measurement properties. The lack of high quality validation data in the meniscal tear population is disappointing given moves to select condition-specific, standardised, 'core' outcome sets for use in clinical trials and general clinical evaluation.[13] Considerable further work is required before this will be possible for patients with meniscal tears.

For the assessment of symptoms and functional status in patients with meniscal tears, there is currently only very limited evidence supporting the selection of the English version of Lysholm or IKDC or Dutch version of KOOS. Although the total score of these three PROMs does not exhibit floor or ceiling effects, a considerable number of subdomain items from both IKDC and Lysholm were reported to have unacceptable floor or ceiling effects. For health-related quality of life, only limited evidence supports the selection of WOMET. One study suggests that measurement error may limit the ability of the WOMET to detect the MIC in score for meniscal patients.[51] Several WOMET subdomain items, but not the total score, have been reported to exhibit unacceptable floor or ceiling effects. For assessment of activity level, only the Tegner Activity Scale has been evaluated, and only very limited evidence is available.

Of all the PROMs evaluated, WOMET has the strongest evidence for content validity. In common with many of the validation studies in this population, however, the included patients frequently had other diagnoses in the same knee such as ligament injuries or chondral defects. This impacts on the interpretation of clinical evidence in subgroups of patients who were poorly represented within the development or validation study population. The findings of these validation studies may not be generalisable to such subgroups, and a PROM may fail to detect important clinical differences. Further validation studies may be required in subgroups or the development of a more specific outcome measure may be necessary.[56] This is pertinent, for example, to current debate about the effectiveness of arthroscopic partial meniscectomy where there is an increasing focus on certain subgroups of patients within this highly heterogeneous population.[12 54 57 58]

### CONCLUSION

Many PROMs have been used in clinical studies of patients with meniscal tears, but the overall quality of evidence supporting the validity of these PROMs is poor. Further work is required, targeting the deficiencies highlighted by this systematic review, to ensure these PROMs truly reflect the symptoms, function and quality of life of patients with meniscal tears. This is necessary to inform the design and interpretation of clinical studies of interventions such as arthroscopic partial meniscectomy in patients with meniscal tears.

**Contributors** SGFA: methodology, study selection, analysis and writing and editing the paper. RM: study selection, analysis and editing the paper. DJB and AJP: concept and editing the paper. SH: methodology, analysis and editing the paper.

**Funding** This report is independent research supported by the National Institute for Health Research (NIHR Doctoral Research Fellowship to SGFA, DRF-2017-10-030).

**Disclaimer** The views expressed in this publication are those of the authors and not necessarily those of the NHS, the National Institute for Health Research or the Department of Health.

**Competing interests** None declared.

**Provenance and peer review** Not commissioned; externally peer reviewed.

**Data sharing statement** No additional data available.

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
