## [Reviewer comments · BMJ Open]

ARTICLE DETAILS

TITLE (PROVISIONAL)	Patient-reported outcome measures for patients with meniscal tears: a systematic review of measurement properties and evaluation with the COSMIN checklist
AUTHORS	Abram, Simon; Middleton, Robert; Beard, David; Price, Andrew; Hopewell, Sally

VERSION 1 – REVIEW

REVIEWER	Benjamin E Smith Derby Teaching Hospitals NHS Foundation Trust, UK; University of Nottingham, UK
REVIEW RETURNED	08-May-2017

GENERAL COMMENTS	The authors have completed a comprehensive and thorough review, and should be commended for their work. I particularly like the summaries of each of the included PROM. They have asked a pertinent question with clinical and research implications. I do have some queries and suggestions for potential improvement. Methods: Little confused by this section. It feels some important information from the PROSPERO protocol is missing from the main manuscript. For example: Condition or domain being studied; Participants/population; Intervention(s), exposure(s). No mention of unpublished trials. Presume they were not searched? Needs to be mentioned as a limitation. No definition of what a meniscal tear is, or on how it should be diagnosed. Heterogeneity mentioned in discussion, and referred to table 2. No mention in the methods how this was assessed. Results: Nice summary of the PROM. Heterogeneity mentioned in discussion, and referred to table 2. No mention in the results. Discussion / Conclusion: I think the paper is missing a research implications section and a clinical implications section. The paper answers an important questions, and I think signposting of the results in relation to research and clinical practice would help readers with the take home messages. The introduction briefly mentions the debate around effectiveness of arthroscopic treatment.
---

	I think this should be expanded to mention the difference (or not) between acute or degenerative symptomatic meniscal tears. e.g.: http://www.bmj.com/content/356/bmj.j356.full and https://www.ncbi.nlm.nih.gov/pubmed/27394143/ I think the discussion section would benefit from mentioning the debate around the prevalence of meniscal tears in symptomatic and asymptomatic. How sure are you that patients on a waiting list for an arthroscopic meniscectomy actually have a symptomatic meniscal tear? e.g. http://www.bmj.com/content/345/bmj.e5339 Some discussion on the diagnostic accuracy of diagnosing meniscal tears may also be of benefit. Where they all confirmed with an MRI? Conclusion: Conclusion section seems a little long. Wonder if middle two paragraphs would work better in the discussion section.
--	--

REVIEWER	Barbara Snoeker Clinical Epidemiology and Biostatistics, University of Amsterdam, The Netherlands
REVIEW RETURNED	23-May-2017

GENERAL COMMENTS	In general At first, I would like to compliment the authors with a well conducted study. It had methodologically strong aspects and was well written. However, I do have major issues that I would like to address. Hopefully, the authors can give clarity about these concerns.  1. Authors argue that PROMS should be validated for patients with meniscal tears. In fact, even in the development phase of the PROMS, meniscal tear patients should be involved. Could the authors explain why this is so important? As authors wrote, an earlier systematic review has been performed on knee osteoarthritis amongst others. How different are PROMS for these patients in comparison to patients with meniscal tears? Which items are specific for the meniscal tear population? 2. Authors also argue that PROMS need to be ideally condition specific. In the introduction section an example has been given for health related quality of life instruments. However, aren't these generic instruments developed for a broad population and diseases? Can't we argue that if these instruments are validated in an orthopaedic knee population, further validation is not necessary? 3. In the discussion section it is written there was heterogeneity in the meniscal tear population. Ligament injury and chondral damage were included. However, is this not just daily clinical practice? Is heterogeneity/variation not necessary for a PROM to be able to measure the full construct? 4. The conclusion states that due to poor quality and incomplete evidence, studies reliant on PROMS should be interpreted with caution. This is a strong conclusion according to the presented evidence. The absence of evidence is not the evidence of absence. As much information about validation of these PROMS is missing, we cannot argue that these PROMS are not suited for meniscal tear patients. What I miss in this article, is the comparison with the
---

PROMS validated in other populations, and the comparability with the meniscal tear population. This manuscript has been written tendentious and is therefore limited interpretable.

Issues per paragraph

Abstract:

As suggested in the previous section, the conclusion is stated too strong based on the results. Results that were described seems limited.

Introduction:

Line 16-17: Why is the interpretation of findings difficult between trials, due to the use of a wide array of PROMS? All outcomes, functional as well as QoL, point in the same direction (no difference between surgical or conservative treatment).

Line 26-27: Why need PROMS to be "as much as possible condition-specific"? Aren't generic instruments not important: quality of patient's life, value based medicine?

Line 29-30: "For a PROM to be valid, it should be developed with condition-specific patient involvement". Can a reference be added? I suggest to rewrite this sentence taking into account the following sentence.

Line 34-35: "Core outcome sets": can this be specified? Does it mean that for every condition a validated questionnaire should be developed?

Line 40-41: The evaluation of measurement properties for PROMS in patients with meniscal tears might be usefull, however the necessity of the research question remains unclear in the introduction.

Methods:

Line 10-11 of paragraph "data extraction":
The system of 'worst score counts' is well explained, however it remains unclear when an item was scored as excellent, good or fair.

Results:

Line 29-30 of paragraph "Quality of PROMS": remove "is" before PROM.

Overall: in the results section multiple times the rating "moderate positive evidence" or "limited positive evidence" has been given. To give insight into the quality of evidence, it is recommended to provide more detailed information in the methods section how this evidence can be interpreted (next to the references that has been given).

	Discussion: Line 36: Internal consistency has been rated poor in most studies with the argument that factor analysis was not performed. It was suggested by the authors that without the assessment of structural validity no interpretation of internal consistency can be made. This is a strict criterium, as reporting of the Cronbach's alpha is a measure of internal consistency. Why do authors hold on to factor analysis? Line 1-12 of the discussion section (page 13): as earlier suggested, it was not clear why PROMS should be developed with meniscal tear patients, and why heterogeneity was a problem. The last sentence of this paragraph: "This makes...arthroscopic meniscectomy" seems preliminary based on the results. Overall: the argument to perform this study is mainly based on the conclusions from the studies investigating arthroscopy compared to conservative treatment for meniscal tears. Authors conclude to interpret these results with caution due to the unknown and poor validation of PROMS for meniscal tear patients. This conclusion is, as earlier prescribed, too strong based on the results. Can it really be argued that the results from several RCTs that conservative treatment should be opted before surgical therapy should be provided, must be interpreted with caution through the unknown and poor validation of PROMS for meniscal tear patients? Conclusion: Overall: the conclusion should be written in one (short) paragraph, and should not include references.
--	---

REVIEWER	Eberbach, Helge Department of Orthopaedic and Trauma Surgery, Freiburg University Hospital, Hugstetter Str. 55, 79106 Freiburg, Germany
REVIEW RETURNED	16-Jun-2017

GENERAL COMMENTS	I believe the premise of the paper is helpful to our readers but several changes are needed to consider publication. The authors evaluate 10 patient-reported outcome measures (PROM) in patients with meniscal tears of the knee. The authors conclude that there is only incomplete evidence for the validity of the current available PROMSs and suggest further validation studies and the development of more specific outcome measures. Positive aspects of the study include the usage of the validated COSMIN checklist for the appraisal of study quality and the uniqueness in the literature. The paper is generally well written with only a few grammar errors. The biggest negatives of the study are that the authors don't explicitly comply to the guidelines for systematic reviews like the PRISMA-guidelines and that the populations of the included studies mainly represent patients with degenerative meniscus tears. The last point should be discussed in detail. Page 2, Line 3: Please correct: Meniscal tears occur frequently... Page 2, Line 3: Please use the term partial meniscectomy in this Line and the rest of the paper. Meniscectomy can be misunderstood as total meniscectomy.
---

	Page 5, Line 57: The quality of each included study... Discussion: Please discuss the mean age (38-53 years) of the patients of the included studies and the following limitations. Figure 1: Adapt the flow chart to the PRISMA guidelines for systematic reviews – study selection.
--	--

VERSION 1 – AUTHOR RESPONSE

Reviewer: 1

Benjamin E Smith

Derby Teaching Hospitals NHS Foundation Trust, UK; University of Nottingham, UK

Comment:

The authors have completed a comprehensive and thorough review, and should be commended for their work. I particularly like the summaries of each of the included PROM. They have asked a pertinent question with clinical and research implications.

Response:

Thank you for these comments.

I do have some queries and suggestions for potential improvement.

Methods.

Comment:

Little confused by this section. It feels some important information from the PROSPERO protocol is missing from the main manuscript. For example: Condition or domain being studied; Participants/ population; Intervention(s), exposure(s).

Response:

Thank you for highlighting this. We have added more detail on these topics and restructured the methods accordingly to enhance clarity.

Comment:

No mention of unpublished trials. Presume they were not searched? Needs to be mentioned as a limitation.

Response:

Added as limitation with reasoning behind this decision.

Comment:

No definition of what a meniscal tear is, or on how it should be diagnosed.

Response:

Thank you. More detail on meniscal tears and diagnosis has been added to the introduction. Some further detail and context in discussion section.

Comment:

Heterogeneity mentioned in discussion, and referred to table 2. No mention in the methods how this was assessed.

Response:

Thank you. Reference to heterogeneity added to generalisability section.

Results.

Comment:

Nice summary of the PROM.

Heterogeneity mentioned in discussion, and referred to table 2. No mention in the results.

Response:

Thank you – this assessment is a descriptive comparison of the included patients in each study in comparison to the population of interest. Table 2 covers each study sample patients in detail. Some specific detail is also included, where relevant, under specific PROMs in the results section. There is further discussion of heterogeneity added to the discussion.

Comment:

Discussion / Conclusion:

I think the paper is missing a research implications section and a clinical implications section. The paper answers an important questions, and I think signposting of the results in relation to research and clinical practice would help readers with the take home messages.

Response:

The discussion and conclusion has been restructured with the intention of making this clearer. Specifically, an 'implications for practice' section has been added.

Comment:

The introduction briefly mentions the debate around effectiveness of arthroscopic treatment. I think this should be expanded to mention the difference (or not) between acute or degenerative symptomatic meniscal tears. e.g.: <http://www.bmj.com/content/356/bmj.j356.full> and <https://www.ncbi.nlm.nih.gov/pubmed/27394143/>

Response:

Thank you – we agree and have added some detail to the discussion, including these references.

Comment:

I think the discussion section would benefit from mentioning the debate around the prevalence of meniscal tears in symptomatic and asymptomatic. How sure are you that patients on a waiting list for an arthroscopic meniscectomy actually have a symptomatic meniscal tear? e.g. <http://www.bmj.com/content/345/bmj.e5339>

Response:

Thank you – more detail on this important issue has been added. Suggested additional reference has been included.

Comment:

Some discussion on the diagnostic accuracy of diagnosing meniscal tears may also be of benefit. Where they all confirmed with an MRI?

Response:

Thank you for this suggestion. Some detail on the inconsistency in the selection of patients has been added.

Conclusion.

Comment:

Conclusion section seems a little long. Wonder if middle two paragraphs would work better in the discussion section.

Response:

Thank you – we agree and have restructured the discussion and conclusion. The conclusion is now a single paragraph.

Reviewer: 2

Barbara Snoeker

Clinical Epidemiology and Biostatistics, University of Amsterdam, The Netherlands

In general

At first, I would like to compliment the authors with a well conducted study. It had methodologically strong aspects and was well written. However, I do have major issues that I would like to address. Hopefully, the authors can give clarity about these concerns.

Thank you for this and the constructive comments that follow. We have responded to each in turn.

Comment:

Authors argue that PROMS should be validated for patients with meniscal tears. In fact, even in the development phase of the PROMS, meniscal tear patients should be involved. Could the authors explain why this is so important? As authors wrote, an earlier systematic review has been performed on knee osteoarthritis amongst others. How different are PROMS for these patients in comparison to patients with meniscal tears? Which items are specific for the meniscal tear population?

Response:

Thank you. We have added more detail to the introduction and discussion regarding the difficulties in distinguishing the symptoms of meniscal tears from those of osteoarthritis and hope that this clarifies the distinction between meniscal tears and osteoarthritis. We have discussed the development and validation of each of the included PROMs in the results section.

Comment:

Authors also argue that PROMS need to be ideally condition specific. In the introduction section an example has been given for health related quality of life instruments. However, aren't these generic instruments developed for a broad population and diseases? Can't we argue that if these instruments are validated in an orthopaedic knee population, further validation is not necessary?

Response:

Thank you – we have extensively revised this section of the introduction to make it clearer. There is a distinction between the purpose of condition specific and generic instruments. Generic instruments, such as the EQ-5D, may enable the comparison of aspects of quality of life, for example, between patients with widely different medical and surgical conditions.

It is clear, however, that such generic instruments cannot be expected to be a comprehensive measure all aspects of quality of life that are important for patients with specific conditions: consider a patient with a traumatic injury of the hand in comparison to a patient with chronic obstructive pulmonary disease. We believe a full discussion of the advantages and disadvantages of generic versus condition specific instruments may be beyond the scope of this review, but have re-written this paragraph of the introduction to make the topic clearer.

Comment:

In the discussion section it is written there was heterogeneity in the meniscal tear population. Ligament injury and chondral damage were included. However, is this not just daily clinical practice? Is heterogeneity/variation not necessary for a PROM to be able to measure the full construct?

Response:

We have added more detail regarding the assessment of patients with meniscal tears and degenerative knee disease to the introduction and discussion. Hopefully this clarifies the importance of distinguishing a symptomatic meniscal tear from osteoarthritis. The fundamental premise of a PROM is that it measures what it intends to measure. This is challenging with meniscal tears, as many may not cause symptoms and frequently occur in patients with osteoarthritis of the knee causing pain. It is essential, however, that if a clinician collects the results of a PROM to measure the success of a treatment such a surgery, the meniscal component of symptoms must be detected by that PROM – both at baseline and follow up. In contrast, there are a number of PROMs such as the Oxford Knee Score that have been designed and validated for use in the population of patients with osteoarthritic knee disease.

Comment:

The conclusion states that due to poor quality and incomplete evidence, studies reliant on PROMS should be interpreted with caution. This is a strong conclusion according to the presented evidence. The absence of evidence is not the evidence of absence. As much information about validation of these PROMS is missing, we cannot argue that these PROMS are not suited for meniscal tear patients. What I miss in this article, is the comparison with the PROMS validated in other populations, and the comparability with the meniscal tear population. This manuscript has been written tendentious and is therefore limited interpretable.

Response:

We have revised the wording of the conclusion to focus only on the PROMs and population. We have referenced reviews that have been performed of the PROMs for use in osteoarthritis and other knee conditions. We have also discussed the original development of each PROM – in many cases this was in patients with osteoarthritis or ligament injury.

Issues per paragraph

Abstract.

Comment:

As suggested in the previous section, the conclusion is stated too strong based on the results. Results that were described seems limited.

Response:

We understand this concern and have revised the conclusion wording to avoid referencing specific studies and summarise only the relevance to the population of patients with meniscal tears.

Introduction.

Comment:

Line 16-17: Why is the interpretation of findings difficult between trials, due to the use of a wide array of PROMS? All outcomes, functional as well as QoL, point in the same direction (no difference between surgical or conservative treatment).

Response:

We have included detail in the introduction regarding the consensus in the clinical trial academic community regarding the importance of consistent use of PROMs (generic and/or condition specific - core outcome sets): <https://www.ncbi.nlm.nih.gov/pubmed/22867278> Discussion of the findings of the 8 clinical trials that have been performed in patients with meniscal tears is, however, beyond the scope of this article.

Comment:

Line 26-27: Why need PROMS to be "as much as possible condition-specific"? Aren't generic instruments not important: quality of patient's life, value based medicine?

Response:

Please see earlier answer and revision to introduction section – we've removed that particular wording to instead reference that PROMs serve different purposes when generic vs condition specific. We agree both may be important and this is the reason that this review did not limit inclusion/exclusion criteria by PROM design (generic, knee-specific, meniscal-specific). All such PROMs are included and described in detail.

Comment:

Line 29-30: "For a PROM to be valid, it should be developed with condition-specific patient involvement". Can a reference be added? I suggest to rewrite this sentence taking into account the following sentence.

Response:

Thank you. We have added reference to the classic paper by Guyatt et. al. which has been used as the basis for the development of disease-specific instruments in healthcare.
<https://www.ncbi.nlm.nih.gov/pubmed/3955482>

Comment:

Line 34-35: "Core outcome sets": can this be specified? Does it mean that for every condition a validated questionnaire should be developed?

Response:

We have reworded these sentences to make it clearer what a core outcome set represents. There is further detail, of course, in the referenced consensus paper.

Comment:

Line 40-41: The evaluation of measurement properties for PROMS in patients with meniscal tears might be useful, however the necessity of the research question remains unclear in the introduction.

Response:

Thank you. We have added more detail to the introduction to, hopefully, make this clearer.

Methods.

Comment:

Line 10-11 of paragraph "data extraction":

The system of 'worst score counts' is well explained, however it remains unclear when an item was scored as excellent, good or fair.

Response:

There are over 100 items across the COSMIN boxes, each with up to four anchor statements for the excellent/good/fair/poor scoring. Describing the scoring detail of each item is therefore, unfortunately, beyond the realistic scope of this article. Hopefully readers will explore the reference (open access) if they are interested in more detail and content of the COSMIN checklist?

<https://www.ncbi.nlm.nih.gov/pubmed/21732199>

Results.

Comment:

Line 29-30 of paragraph "Quality of PROMS": remove "is" before PROM.

Response:

Thank you – corrected.

Comment:

Overall, in the results section multiple times the rating "moderate positive evidence" or "limited positive evidence" has been given. To give insight into the quality of evidence, it is recommended to provide more detailed information in the methods section how this evidence can be interpreted (next to the references that has been given).

Response:

A summary of the meaning of these terms is included in Table 1 and referenced in the data synthesis sections.

Discussion.

Comment:

Line 36: Internal consistency has been rated poor in most studies with the argument that factor analysis was not performed. It was suggested by the authors that without the assessment of structural validity no interpretation of internal consistency can be made. This is a strict criterium, as reporting of the Cronbach's alpha is a measure of internal consistency. Why do authors hold on to factor analysis?

Response:

The importance of factor analysis was decided by international consensus Delphi panel when COSMIN was developed. The included reference (open access) describes the rationale behind this decision in detail (<https://www.ncbi.nlm.nih.gov/pubmed/20298572>). Briefly, however, as written in the discussion, the unidimensionality of the PROM or PROM sub-domain (structural validity) must be confirmed by factor analysis before the internal consistency statistic can be interpreted.

Comment:

Line 1-12 of the discussion section (page 13): as earlier suggested, it was not clear why PROMS should be developed with meniscal tear patients, and why heterogeneity was a problem. The last sentence of this paragraph: "This makes...arthroscopic meniscectomy" seems preliminary based on the results.

Response:

This section has been heavily revised to enhance clarity and highlight the clinical relevance and to address these comments.

Comment:

Overall, the argument to perform this study is mainly based on the conclusions from the studies investigating arthroscopy compared to conservative treatment for meniscal tears. Authors conclude to interpret these results with caution due to the unknown and poor validation of PROMS for meniscal tear patients. This conclusion is, as earlier prescribed, too strong based on the results. Can it really be argued that the results from several RCTs that conservative treatment should be opted before surgical therapy should be provided, must be interpreted with caution through the unknown and poor validation of PROMS for meniscal tear patients?

Response:

Thank you – this has been revised to remove references to other studies and keep the clinical relevance to the population studied (patients with meniscal tears).

Conclusion.

Comment:

Overall, the conclusion should be written in one (short) paragraph, and should not include references.

Response:

Thank you – this has been revised.

Reviewer: 3

Eberbach, Helge

Department of Orthopaedic and Trauma Surgery, Freiburg University Hospital, Hugstetter Str. 55, 79106 Freiburg, Germany

Comment:

I believe the premise of the paper is helpful to our readers but several changes are needed to consider publication. The authors evaluate 10 patient-reported outcome measures (PROM) in patients with meniscal tears of the knee. The authors conclude that there is only incomplete evidence for the validity of the current available PROMSs and suggest further validation studies and the development of more specific outcome measures.

Response:

Thank you.

Comment:

Positive aspects of the study include the usage of the validated COSMIN checklist for the appraisal of study quality and the uniqueness in the literature. The paper is generally well written with only a few grammar errors. The biggest negatives of the study are that the authors don't explicitly comply to the guidelines for systematic reviews like the PRISMA-guidelines and that the populations of the included studies mainly represent patients with degenerative meniscus tears. The last point should be discussed in detail.

Response:

Thank you for this feedback. We have revised the manuscript to more closely follow the PRISMA structure. A reference to PRISMA has been added at the start of the methods. The PRISMA flow chart has been revised. Regarding the degenerative meniscal tear patient population, in response to these comments we have added detail on this topic to the discussion, and also included mean age details in the study characteristics section of the results.

Comment:

Page 2, Line 3: Please correct: Meniscal tears occur frequently...

Response:

Thank you – corrected.

Comment:

Page 2, Line 3: Please use the term partial meniscectomy in this Line and the rest of the paper. Meniscectomy can be misunderstood as total meniscectomy.

Response:

We agree – standardised throughout.

Comment:

Page 5, Line 57: The quality of each included study...

Response:

Thank you – corrected.

Discussion.

Comment:

Please discuss the mean age (38-53 years) of the patients of the included studies and the following limitations.

Response:

Thank you for this suggestion – we have added a paragraph discussing this topic to the discussion. The age of the patients has also been added within the study characteristics section of the methods.

Comment:

Figure 1: Adapt the flow chart to the PRISMA guidelines for systematic reviews – study selection.

Response:

Thank you – this has been revised.

VERSION 2 – REVIEW

REVIEWER	Barbara Snoeker University of Amsterdam, The Netherlands
REVIEW RETURNED	02-Aug-2017

GENERAL COMMENTS	Overall: Compliments to the authors for their revision. Clarity of the main message has improved. It now states the true issue on this topic with the focus only on the PROMs for meniscal tear patients. I have only minor suggestions: Introduction: “Meniscal tears are diagnosed and managed based upon a combination of a review of symptoms, clinical examination, and imaging findings on x-ray radiographs and magnetic resonance imaging (MRI).” → meniscal tears are not diagnosed on X-ray radiographs. “This inconsistency leads to restricted comparisons between trials and difficult interpretation of their findings” → I agree with the restricted comparisons, however as all outcomes point in the same direction, I don’t agree that it is difficult to interpret the findings in this specific situation (APM effectiveness versus conservative treatment). My suggestion is to remove the second part of this sentence. Discussion: The problem of making a distinction between meniscal and osteoarthritic pain has been well stated by the authors. The discussion section is much improved. Conclusion: Remove “In summary,” in the first sentence.
---

VERSION 2 – AUTHOR RESPONSES

Reviewer: 2

Reviewer Name: Barbara Snoeker

Institution and Country: University of Amsterdam, The Netherlands

Please state any competing interests or state 'None declared': None declared

Please leave your comments for the authors below

Comment:

Overall, Compliments to the authors for their revision. Clarity of the main message has improved. It now states the true issue on this topic with the focus only on the PROMs for meniscal tear patients. I have only minor suggestions:

Response:

Thank you.

Introduction.

Comment:

“Meniscal tears are diagnosed and managed based upon a combination of a review of symptoms, clinical examination, and imaging findings on x-ray radiographs and magnetic resonance imaging (MRI).” → meniscal tears are not diagnosed on X-ray radiographs.

Response:

Thank you, we have corrected this sentence, removing the reference to x-ray and imaging modalities.

Comment:

“This inconsistency leads to restricted comparisons between trials and difficult interpretation of their findings” → I agree with the restricted comparisons, however as all outcomes point in the same direction, I don't agree that it is difficult to interpret the findings in this specific situation (APM effectiveness versus conservative treatment). My suggestion is to remove the second part of this sentence.

Response:

We have removed the second part of the sentence as requested. Thank you.

Comment:

Discussion:

Response:

The problem of making a distinction between meniscal and osteoarthritic pain has been well stated by the authors. The discussion section is much improved.

Response:

Thank you.

Conclusion.

Comment:

Remove “In summary,” in the first sentence.

Response:

Removed.